# Epidemiology and trends of animal bites in Khoy County, Northwest Iran: A retrospective study (2021–2024)

Parviz Shahmirzalou[1], Hassan Ebrahimpour Sadagheyani[2,3]*

1 Department of Public Health, Khoy University of Medical Sciences, Khoy, Iran, 2 Department of Health Information Technology, Khoy University of Medical Sciences, Khoy, Iran, 3 Department of Health Information Technology, Neyshabur University of Medical Sciences, Neyshabur, Iran

* sadageyani@yahoo.com

## Abstract

### Background

Animal bites are a significant global public health concern because they can lead to rabies, a disease with a very high case fatality. Therefore, this study aimed to describe the demographic and epidemiological characteristics of animal bite cases and to examine temporal trends in their incidence in Khoy County, Northwest Iran, from 21/03/2021–19/03/2024.

### Methods

This registry-based, retrospective longitudinal study included all animal bite cases recorded in Khoy County between 21/03/2021 and 19/03/2024. Data on demographic characteristics, post-exposure preventive actions, and vaccination status were extracted. Analyses comprised descriptive statistics, chi-square tests in SPSS V17, trend analysis, time-series regression, and a 12-month forecast in Minitab V22.

### Results

The mean age of bite victims was 31.23 years. The incidence of animal bites per 100,000 population in 2021–2024 was 371.39, 503.23, 506.02, and 397.45, respectively. Dogs accounted for 91% of bite incidents, and the most common circumstances leading to bites were sudden animal attacks, provoking the animal, hunting, and self-defense. Monthly counts revealed a clear 12-month seasonal variation, with the highest incidence occurring in summer and early autumn. The forecasting model predicted that late spring and summer of 2024 would be the peak periods for animal-bite incidence.

**Data availability statement:** The data underlying this study are owned by Khoy University of Medical Sciences (Khoy, Iran) and contain potentially sensitive health information. Due to institutional data ownership policies and privacy considerations, the dataset cannot be made publicly available. Qualified researchers may request access to anonymized and de-identified data. Data requests will be reviewed by the Research Committee of Khoy University of Medical Sciences. Requests should be directed to: Dr. Mobin Sokhanvar Head of the Research Committee Khoy University of Medical Sciences Email: Sokhanvar_m@khoyums.ac.ir.

**Funding:** The author(s) received no specific funding for this work.

**Competing interests:** The authors have declared that no competing interests exist.

## Conclusion

Due to the high rate of animal bites in Khoy County, it is advisable to focus on educating at-risk groups about prevention and encouraging prompt medical attention for post-exposure rabies prophylaxis when bites happen.

## Introduction

Animal bites constitute a major public health concern worldwide because, if untreated, they can result in rabies, cause physical injury, and impose substantial healthcare costs [1]. Rabies is a lethal viral disease that affects the central nervous system of mammals, including humans; animal bites are a primary route of transmission. Although rabies is vaccine-preventable, it remains a significant public health problem, particularly in areas with limited access to vaccines and post-exposure prophylaxis. Globally, rabies kills tens of thousands of people annually, the majority of whom live in Asia and Africa [2].

According to Iran's national guidelines for rabies prevention, bites by domestic or wild animals such as dogs, foxes, wolves, jackals, bears, hyenas, cats, bats, raccoons, martens (including species referred to locally as marmot, Indian Grey Mongoose, stoat), ferrets, badgers, and leopards may place a person at risk of rabies; therefore, exposed persons should receive appropriate post-exposure prophylaxis [3]. In Iran, the main source of human rabies is animal bites, primarily from dogs and wild animals such as foxes and wolves [4]. Given the high case fatality rate associated with clinical rabies, early prevention and control are critical. Rabies is a twin concern in Iran, threatening both public health and agriculture, and the economic burden of treatment expenses and livestock losses highlights the need for epidemiological investigations.

The U.S. Centers for Disease Control and Prevention reports that rabies causes approximately 70,000 deaths annually worldwide and that 99% of human rabies cases outside the United States are attributable to dogs; the principal drivers of rabies are inadequate vaccination coverage and limited access to treatment services in some countries [5].

The incidence of animal bites in Iran varies substantially by geographic location and city. The highest reported incidences (per 100,000 population) include Chaldoran in West Azarbaijan with 541 cases [6], Galikesh in Golestan with 540 cases [7], and Bardsir in Kerman with 433 cases [8]. The lowest incidences have been reported in cities such as Sanandaj in Kurdistan province (67 cases) [9], Shiraz in Fars province (154.4 cases) [10], and Kermann (295 cases) [11].

Khoy County covers an area of 5,548 km² in northern West Azarbaijan Province (Fig 1). The county is situated within high mountain ranges and experiences a range of climatic conditions, from warm summers to frigid winters. Parts of the county are forested and host diverse flora and wildlife, including brown bears, gazelles, wild sheep, wild boar, hyenas, and raptors such as eagles and falcons [12].

Although animal bites are known to pose a risk of disease, there is limited information on the epidemiology, temporal trends, and geographic distribution of bites

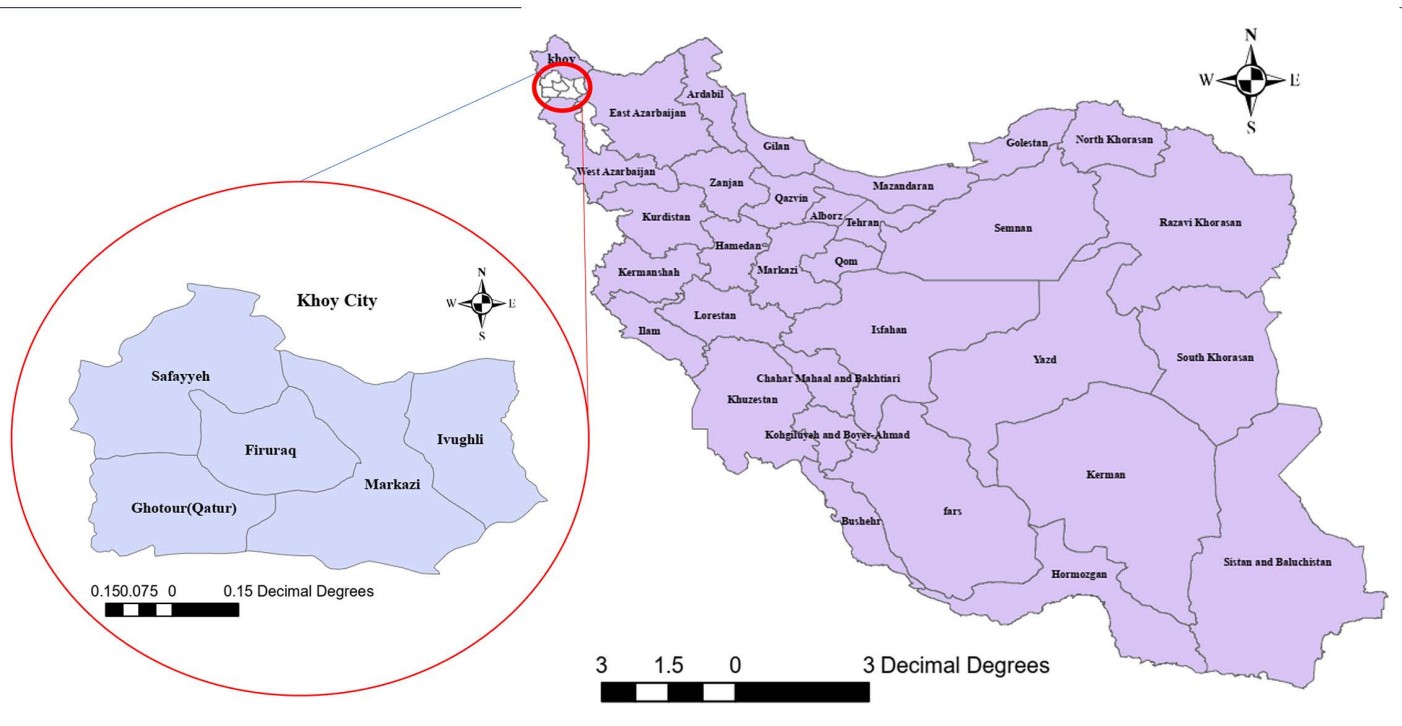

**Fig 1. Geographical location of Khoy County in West Azerbaijan Province, Iran.** The figure was created using ArcGIS software version 10.8 by the authors.

in Khoy County. An informal report from the local rabies prevention center in Khoy stated that no confirmed rabies cases had been reported in recent years. Still, a confirmed human rabies case was reported this year. Although national and provincial statistics provide a general overview, county-level studies that incorporate local temporal and spatial character-istics can inform public health interventions more precisely, and relatively few studies have examined demographic and environmental factors that contribute to bite rates over time [13,14]. The lack of such knowledge impacts policymakers, healthcare providers, and local authorities, who need accurate local data to design prevention and education programs to decrease bite risk. Therefore, this study was designed to describe demographic and epidemiologic features of animal bites in Khoy County, western Iran.

## Materials and methods

This registry-based, retrospective longitudinal study included all individuals who presented to the Animal Bite Prevention Center at Khoy University of Medical Sciences following an animal bite between 21/03/2021 and 19/03/2024. A com-prehensive dataset from this registry was accessed and analyzed for the present study on 26/03/2025. According to the national guideline for animal-bite prevention, any contact between domestic or wild animals and humans involving the ani-mal's teeth and resulting in scratches, bruises, or skin laceration is defined as an animal bite. Animals listed in this guide-line include dogs, foxes, wolves, jackals, bears, hyenas, cats, bats, raccoons, martens (marmot, Indian Grey Mongoose, stoat), ferrets, badgers, and leopards [3].

## Data collection instrument

This study was supported and conducted under a research project approved by the Khoy University of Medical Sciences (Ethics Code: IR.KHOY.REC.1404.001:Link). Khoy County had five districts (Markazi, Ivughli, Firuraq, Safayyeh, Ghotour (or Qatur)) with a population of 348,664 people according to the 2016 (1395 Jalali-Hijri Shamsi calendar) national census, and 198,845 living in urban areas. Unfortunately, the exact population for 2021–2024 was unavailable, so this study estimated it using data obtained from the Vice-Chancellor of Health at Khoy University of Medical Sciences. The population of Khoy County and its districts matched with the Khoy County Governing Office.

Study participants were citizens of Khoy County who had been bitten by an animal between 2021 and 2024, and were registered at the national Centers for Disease Control and Prevention (NCDCP). Participants who met the specified criteria were included in the study. According to the national guideline for animal-bite prevention, individuals with a new bite and three months after the last vaccination are classified as a new case and vaccinated at 0 and 3 days after the animal bite. Individuals' data were collected using a researcher-designed checklist based on records from the National Centers for Disease Control and Prevention (NCDPC). The checklist included:

• Demographic characteristics of the victims;

• Vaccination details, including number and timing of doses;

• Serum administration and wound management information;

• Site and nature of the injury;

• Type of animal (domestic/wild, owned/stray, or under observation).

• The authors did not have access to the participants' identifying information.

## Occupational grouping

• Group 1 includes children, schoolchildren, students, and housewives.

• Group 2 includes animal-related jobs, including rangers, farmers, ranchers, keepers of livestock, veterinarians, woodcutters, and shepherds.

• Group 3: non-animal-related occupations, including military personnel, office workers, laborers, staff of care facilities, janitors, and drivers.

## Classification of bite circumstances

Reasons for animal bites were categorized into four groups:

1. Sudden animal attack, provoking the animal, hunting, or self-defense;

2. Feeding, playing with, or caring for an animal;

3. Contact with a sick or rabid animal;

4. Other reasons.

## Statistical analysis

Both descriptive statistics (measures of central tendency and dispersion, and graphical representations) and inferential statistics (Chi-square or Fisher's exact test, as appropriate, and Z test to compare proportions) were used for data

analysis by SPSS (Statistical Package for Social Sciences) Software Version 17. Time-series analysis in Minitab Version 22 was used to examine the monthly animal bites over time and forecast the number of bites for the next 12 months.

1. Tested for autocorrelation: Correlation across time was assessed using autocorrelation (ACF) and partial autocorrelation function (PACF) plots.

2. Tested for stationarity: Mean and variance stationarity were tested for using trend analysis and the Augmented Dickey–Fuller (ADF) test, and when non-stationarity was found, data transformation and differencing were used.

3. Fitted a time-series regression model of the ARIMA or SARIMA family: The difference parameters (d and D) were set to zero initially. The model was fitted after achieving stationarity.

4. The best model had the minimum Akaike Information Criteria corrected (AICc) and also had no violation from regression assumptions about residuals (normal distribution, randomness, and stability in variance) [15].

## Results

According to the findings of the present study, a total of 6,543 cases of animal bites were reported in Khoy County from 21/03/2021–19/03/2024 – specifically 1,337, 1,840, 1,878, and 1,488 cases in each respective year. The annual incidence rates per 100,000 population were 371.39, 503.23, 506.02, and 397.45 for the years 2021, 2022, 2023, and 2024, respectively. The mean±SD age of bite victims was 31.23±18.19 years, 31.15±20.44 years among females, and 31.25±17.57 years among males. Tables 1, 2 describe animal bite cases from demographic, geographical, and clinical aspects. To check the equality of proportions within subgroups of a feature, the Z test has been used. All comparisons reported significant differences in the distribution of animal bites across subgroups. For example, 79.5% of cases involved males, and dogs accounted for 91.1% of animal bites. Similar results are reported in Table 2.

### Post-exposure prophylaxis and wound management

Between 2021 and 2024, 98%, 97%, 96%, and 97% of bite victims, respectively, sought timely post-exposure prophylaxis (PEP). Overall, 12% received 2 vaccine doses, 86% received 3 doses, and 3% received 4 doses. According to the national guideline for animal-bite prevention, vaccines are injected intradermally. In 98% of all cases, the wound site was washed with soap and water immediately after exposure. The number of male victims was consistently higher than that of females, and this difference was statistically significant (P<0.001). But the proportion of male and female victims did not vary significantly over the years studied. Group 1 (children, students, university students, and housewives) had the highest proportion of bite cases between jobs, and the proportions across occupational categories were significantly different (P<0.001).

The maximum number of bites was in:

- Group 2 in 2021,

- Group 1 in 2022 and 2023,

- Group 3 in 2024. In 2021,

    Overal, 60.3% of bites occurred in rural areas, whereas in 2024, 50.0% were reported in urban areas.

### Cause and type of animals involved

The most common causes of bites were sudden animal attacks, provocation, hunting, defense, feeding, playing with, or caring for animals. Throughout the study period, dogs and domestic animals were the predominant sources of bites, and 73.4% of the biting animals were owned or under observation. Antibiotics were prescribed in 5.4% of cases, and anti-rabies serum was administered in 10.7%.

**Table 1. A description of animal bites based on demographic features from 2021 to 2024, and also checking for an equal proportion of subgroups for every feature.**

| Feature(subgroups) | | Frequency (%) | | | | | Statistic(P) |
|---|---|---|---|---|---|---|---|
| | | 2021 | 2022 | 2023 | 2024 | Total | |
| Gender | Female | 155(11.6%) | 364(19.8%) | 456(24.3%) | 364(24.5%) | 1339(20.5%) | Z=47.80 (P<0.001) |
| | Male | 1182 (88.4%) | 1476(80.2%) | 1422(75.7%) | 1124(75.5%) | 5204(79.5%) | |
| Job | Group One | 344 (25.7%) | 822 (44.7%) | 899 (47.9%) | 528 (35.5%) | 2593 (39.6%) | Z=−24.31 (P<0.001) |
| | Group Two | 741 (55.4%) | 316 (17.2%) | 243 (12.9%) | 327 (22.0%) | 1627(24.9%) | |
| | Group Three | 245 (18.3%) | 689 (37.4%) | 700 (37.3%) | 631 (42.4%) | 2265(34.6%) | |
| | Missing | 7(0.5%) | 13(0.7%) | 36(1.9%) | 2(0.1%) | 58 (.9%) | |
| Cause of Animal Bite | Reason 1 | 1100(82.3%) | 1350(73.4%) | 1390(74.0%) | 982(66.0%) | 4822(73.7%) | Z=−78.55 (P<0.001) |
| | Reason 2 | 197(14.7%) | 449(24.4%) | 461(24.5%) | 235(15.8%) | 1342(20.5%) | |
| | Reason 3 | 2(0.1%) | 11(0.6%) | 18(1.0%) | 233(15.7%) | 264(4.0%) | |
| | Reason 4 | 37(2.8%) | 28(1.5%) | 8(0.4%) | 18(1.2%) | 91(1.4%) | |
| | Missing | 1(0.1%) | 2(0.1%) | 1(0.1%) | 20(1.3%) | 24(0.4%) | |
| Animal Type | Dog | 1333(99.7%) | 1826(99.2%) | 1866(99.4%) | 935(62.8%) | 5960(91.1%) | Z=−80.14 (P<0.001) |
| | Cat | 3(0.2%) | 12(0.7%) | 6(0.3%) | 472(31.7%) | 493(7.5%) | |
| | Goat and Sheep | 0(0%) | 0(0%) | 0(0%) | 34(2.3%) | 34(0.5%) | |
| | Other | 0(0%) | 0(0%) | 0(0%) | 25(1.7%) | 25 (0.4%) | |
| | Missing | 1(0.1%) | 2(0.1%) | 6(0.3%) | 22(1.5%) | 31(0.5%) | |
| Animal Status Before Bite | Domestic | 1229(91.9%) | 1506(81.8%) | 1614(85.9%) | 1180(79.3%) | 5529(84.5%) | Z=−64.93 (P<0.001) |
| | Wild | 2(0.1%) | 8(0.4%) | 10(0.5%) | 12(.8%) | 32(0.5%) | |
| | Stray | 6(0.4%) | 169(9.2%) | 100(5.3%) | 232(15.6%) | 507(7.7%) | |
| | Missing | 100 (7.5%) | 157 (8.5%) | 154 (8.2%) | 64 (4.3%) | 475 (7.3%) | |
| Animal Status After Bite | Owned or supervised | 1207(90.3%) | 1458(79.2%) | 1433(76.3%) | 704(47.3%) | 4802(73.4%) | Z=−73.77 (P<0.001) |
| | Escaped | 114(8.5%) | 242(13.2%) | 218(11.6%) | 230(15.5%) | 804(12.3%) | |
| | Killed | 4(0.3%) | 16(0.9%) | 21(1.1%) | 17(1.1%) | 58(0.9%) | |
| | Missing | 12(0.9%) | 124(6.7%) | 206(11.0%) | 537(36.1%) | 879(13.4%) | |
| Bite by Animal Owner | No | 399(29.8%) | 979(53.2%) | 1366(72.7%) | 1074(72.2%) | 3818(58.4%) | Z=−13.99 (P<0.001) |
| | Yes | 937(70.1%) | 859(46.7%) | 508(27.1%) | 394(26.5%) | 2698(41.2%) | |
| | Missing | 1(0.1%) | 2(0.1%) | 4(0.2%) | 20(1.3%) | 27(.4%) | |

## Temporal distribution

The lowest monthly numbers of bites during 2021–2024 occurred in:• Farvardin (March–April) – 136 cases,• Bahman (January–February) – 130 cases,• Dey (December–January) – 130 cases,• Azar (November–December) – 134 cases. The highest numbers were recorded in:

- Shahrivar (August–September) – 183 cases,

- Mordad (July–August) – 196 cases,

- Shahrivar (August–September) – 185 cases,

- Mehr (September–October) – 185 cases (Fig 2).

## Seasonal pattern and trend analysis

The average number of bites per month in Fig 3 shows that mid-year months had the highest incidence rates.

**Table 2. A description of animal bites based on clinical and geographical features from 2021 to 2024, and also checking an equal proportion of subgroups for every feature.**

| Feature(subgroups) | | Frequency (%) | | | | | Statistic(P) |
|---|---|---|---|---|---|---|---|
| | | 2021 | 2022 | 2023 | 2024 | Total | |
| Antibiotic Prescription | No | 1253(93.7%) | 1701(92.4%) | 1797(95.7%) | 1437(96.6%) | 6188(94.6%) | Z=−72.22 (P<0.001) |
| | Yes | 84(6.3%) | 139(7.6%) | 79(4.2%) | 51(3.4%) | 353(5.4%) | |
| | Missing | 0(0%) | 0(0%) | 2(0.1%) | 0(0%) | 2(0.0%) | |
| Vaccine Type | PCEC | 1(0.1%) | 2(.1%) | 278(14.8%) | 43(2.9%) | 324(5.0%) | Z=72.63 (P<0.001) |
| | VERO | 1335(99.9%) | 1835(99.7%) | 1595(84.9%) | 1441(96.8%) | 6206(94.8%) | |
| | Missing | 1(0.1%) | 3(0.2%) | 5(0.3%) | 4(0.3%) | 13(0.2%) | |
| Serum Therapy | No | 1256(93.9%) | 1590(86.4%) | 1634(87.0%) | 1359(91.3%) | 5839(89.2%) | Z=−63.57 (P<0.001) |
| | Yes | 81(6.1%) | 250(13.6%) | 243(12.9%) | 129(8.7%) | 703(10.7%) | |
| | Missing | 0(0%) | 0(0%) | 1(0.1%) | 0(0%) | 1(0.1%) | |
| Residence Place | Village | 1033(77.3%) | 986(53.6%) | 984(52.4%) | 704(47.3%) | 3707(56.7%) | Z=−11.72 (P<0.001) |
| | City | 296(22.1%) | 821(44.6%) | 888(47.3%) | 751(50.5%) | 2756(42.1%) | |
| | Missing | 8(0.6%) | 33(1.8%) | 6(0.3%) | 33(2.2%) | 80(1.2%) | |
| Incident Position | At Home or Work | 1330(99.5%) | 1820(98.9%) | 1849(98.5%) | 1472(98.9%) | 6471(98.9%) | Z=43.06 (P<0.001) |
| | While traveling | 7(0.5%) | 16(0.8%) | 24(1.3%) | 12(0.8%) | 59(0.9%) | |
| | Missing | 0(0%) | 4(0.2%) | 5(0.3%) | 4(0.3%) | 13(0.2%) | |
| Incident Area | Village | 1074(80.3%) | 1082(58.8%) | 1046(55.7%) | 744(50.0%) | 3946(60.3%) | Z=−18.13 (P<0.001) |
| | City | 247 (18.5%) | 709 (38.5%) | 804 (42.8%) | 723 (48.6%) | 2483(37.9%) | |
| | Missing | 16(1.2%) | 49(2.7%) | 28(1.5%) | 21(1.4%) | 114(1.8%) | |

Both the autocorrelation and partial autocorrelation plots in Fig 4 showed that the first lag was significant, meaning that the number of bites in a given month was significantly related to the number of bites in the previous month.

Subsequent trend analysis yielded the following regression equation:

$$\text{Equation 1}: \text{"Number of bite cases"} = 162.92 - 0.25 \times \text{"Time period"}$$

The regression coefficient (–0.25) was not statistically significant (P=0.186), as shown in Equation (1), which showed no significant linear trend in the monthly frequency of bites. The Augmented Dickey–Fuller (ADF) test statistic was "–3.55" (P=0.007), indicating the data were stationary.

### Time-series model and forecast

Since the mean and variance were stationary, time-series modeling was conducted. In total, 20 models with different components were fitted. Table 3 reports the goodness-of-fit indices.

The final fitted model was SARIMA(2,0,0)(1,0,0)$_{12}$, with an autoregressive component of order 3, a first-order seasonal autoregressive component, and a 12-month seasonal period. The model parameters are shown in Table 4. According to Equation (1), the regression coefficient (–0.25) was not statistically significant (p=0.186), indicating that no meaningful linear trend existed in the monthly frequency of bites.

The Box–Pierce (Ljung–Box) chi-square test for lags 12, 24, and 36 was not significant (P=0.497, 0.413, 0.610), indicating no autocorrelation among residuals. Plots related to residual assumptions are included in the Supporting Information (S1 Fig–S3 Fig). Based on this final model, the monthly number of animal-bite cases for the subsequent 12 months was predicted (Table 4).

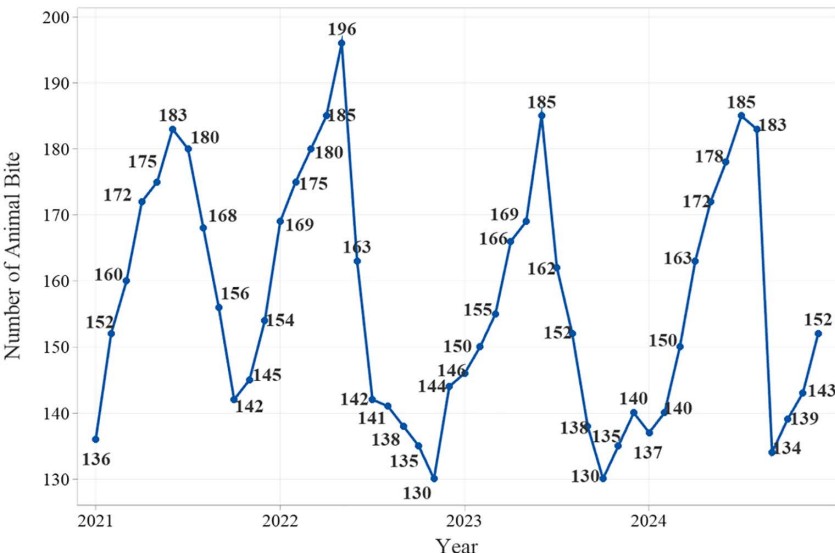

**Fig 2. Number of animal bite cases in Khoy County from 2021 to 2024.**

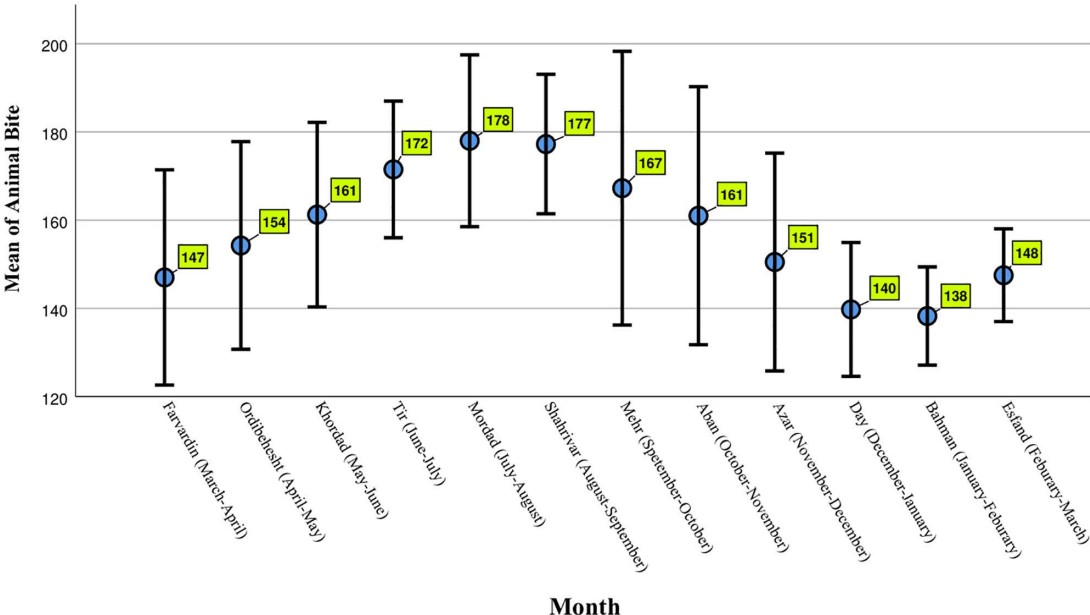

**Fig 3. Monthly incidence of animal bites, along with 95% confidence interval for the mean animal bites in that month.**

The predicted number of cases is expected to increase initially, decrease, and then increase again toward the end of the forecast year (see Fig 5), reflecting a seasonal pattern.

## Discussion

The present study aimed to investigate the epidemiological characteristics and temporal variations of animal-bite incidence in Khoy County between 2021 and 2024. Most bite victims belong to occupational group 1, which includes children,

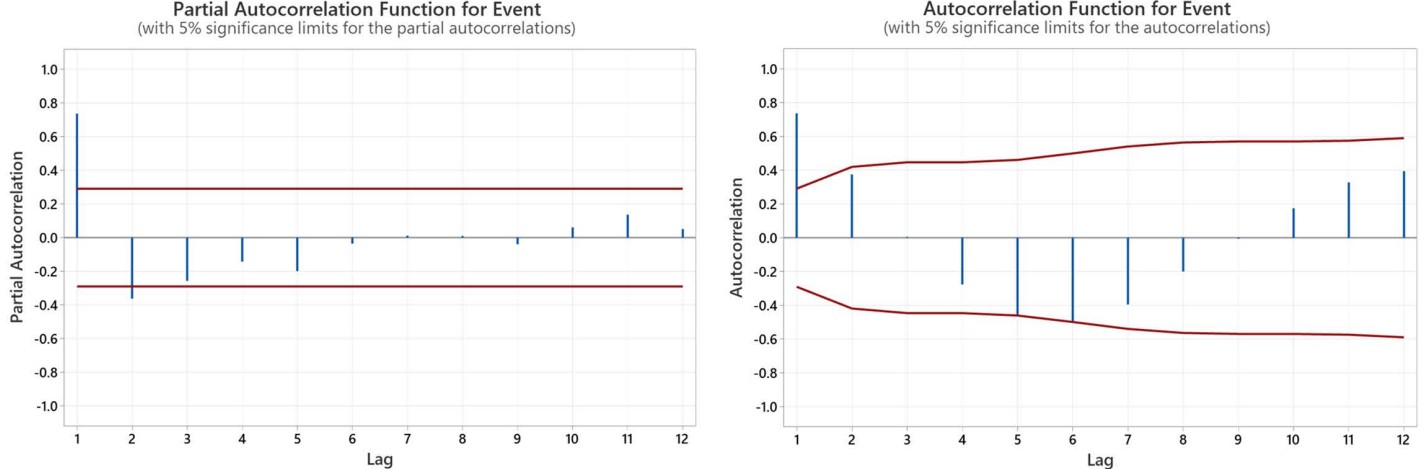

**Fig 4. Autocorrelation (right) and partial autocorrelation (left) graphs of the number of animal bites during the years 2021 to 2024.**

**Table 3. Goodness of fit indices for models. The differentiation component is set to zero for all models.**

| ID | Model (d = 0, D = 0) | LogLikelihood | AICc | AIC | BIC |
|---|---|---|---|---|---|
| 1 | p = 3, q = 2, P = 3, Q = 0 | 13512 | −26998 | −27004 | −26986 |
| 2 | p = 2, q = 0, P = 1, Q = 0 | 6769 | −13511 | −13517 | −13499 |
| 3 | p = 3, q = 0, P = 3, Q = 0 | 3775 | −7530 | −7534 | −7519 |
| 4 | p = 3, q = 1, P = 3, Q = 2 | 3031 | −6037 | −6043 | −6024 |
| 5 | p = 0, q = 0, P = 2, Q = 2 | 467 | −922 | −924 | −914 |
| 6 | p = 2, q = 4, P = 2, Q = 0 | 145 | −267 | −272 | −255 |
| 7 | p = 0, q = 0, P = 3, Q = 3 | −46 | 108 | 105 | 118 |
| 8 | p = 2, q = 2, P = 2, Q = 2 | −69 | 165 | 159 | 178 |
| 9 | p = 3, q = 3, P = 2, Q = 0 | −104 | 234 | 228 | 246 |
| 9 | p = 4, q = 1, P = 2, Q = 0 | −136 | 295 | 291 | 307 |
| 10 | p = 1, q = 1, P = 2, Q = 1 | −152 | 320 | 317 | 331 |
| 11 | p = 1, q = 4, P = 2, Q = 0 | −163 | 348 | 343 | 360 |
| 12 | p = 2, q = 1, P = 0, Q = 0 | −171 | 354 | 353 | 362 |
| 13 | p = 3, q = 0, P = 0, Q = 0 | −172 | 356 | 355 | 364 |
| 14 | p = 2, q = 0, P = 0, Q = 0 | −174 | 357 | 356 | 363 |
| 15 | p = 2, q = 2, P = 0, Q = 0 | −171 | 357 | 355 | 366 |
| 16 | p = 3, q = 1, P = 0, Q = 0 | −171 | 357 | 355 | 366 |
| 17 | p = 3, q = 0, P = 0, Q = 1 | −172 | 358 | 356 | 367 |
| 18 | p = 4, q = 0, P = 0, Q = 0 | −172 | 358 | 356 | 367 |
| 19 | p = 5, q = 0, P = 0, Q = 0 | −171 | 358 | 355 | 369 |
| 20 | p = 2, q = 0, P = 0, Q = 1 | −173 | 358 | 357 | 366 |

students, university students, and housewives. There is a significant difference in the proportion of subgroups, especially for the mentioned features.

Similar to the present findings, Rostampour [16], Khan [17], Kiakalayeh [4], and Varisli [18], also reported that the majority of bite victims were male. In contrast, a study conducted in Greece found that 41% of dog-bite victims were

Table 4. The final model for the time series of animal bites in Khoy County.

| Parameter | | Beta | t (P-value) |
|---|---|---|---|
| Constants | | 59.5 | 60.65 (<0.001) |
| Autocorrelation | 1st order | 1.19 | 7.96 (0.001) |
| | 2nd order | −0.30 | −1.30 (0.200) |
| | 3rd order | −0.27 | −1.77 (0.083) |
| Seasonal Moving average-12 | | 0.23 | 1.29 (0.202) |

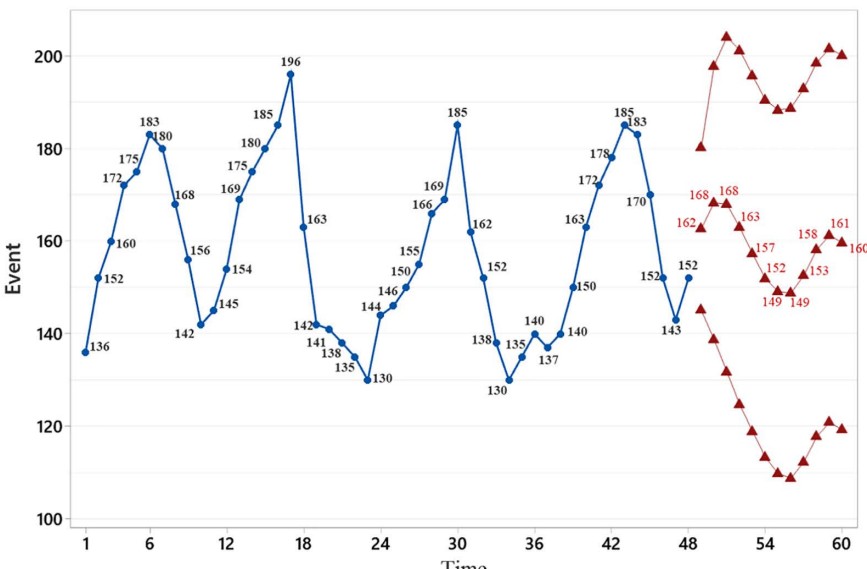

**Fig 5. Monthly incidence of animal bites in Khoy County in 2021-2024, along with prediction of the number of cases in the next 12 months using a regression model.**

female [19], which is considerably higher than in our study (21%). Mirzaei et al.[20], in East Azerbaijan Province, found that most victims were workers, farmers, or livestock breeders, whereas Alian et al.[21] reported that "self-employed" individuals were most frequently affected.

In the current study, the bite site varied, unlike previous findings. For example, Khan et al.[17] reported that 70% of bites in children and 86% in adults occurred on the lower extremities. In contrast, the present study found these proportions to be 46% and 38%, respectively, indicating a lower rate of leg injuries. The mean age of bite victims in our study was 31.23 years, with the 10–19 and 40–49 age groups most frequently affected.

Rostampour [16] reported that the 19–59-year-old age group (62%) was most affected; Kiakalayeh [4] found the "more than 50" age group (24%) to be dominant, and Khan et al. [17] observed that 61% of victims were over 18 years old. Similarly, Dougas [19] reported that individuals aged 50–59 years old had the highest incidence, while Varisli [18] found mean ages of 22 years for men and 26 years for women.

Regarding the circumstances of bites, the most common reasons in the study were provoking the animal, sudden attacks, or self-defense. Although dogs were responsible for the vast majority (91.1%) of bite cases—mainly due to their proximity to humans—further analysis showed that cat bites had increased sharply in recent years, rising from 0.2% in 2021 to 32% in 2024.

This increase may reflect a growing population of stray cats and dogs in urban areas. In most cases, the attacking animal was owned or under observation, and in 10–47% of these cases, the animal bit its own owner. Overall, approximately 60% of animal-bite cases occurred in village areas, similar to the report by Varisli [18]. In a study conducted in Isfahan, Abbaspour et al. [22] reported that the leading cause of bites was sudden animal attack (49%). Among males, the most common reason was sudden attack (52%), while among females it was self-defense (17%).

In our study, most victims received the VERO cell vaccine, and in nearly 94.6% of cases, neither antibiotics nor anti-rabies serum were required. Nikbakht et al.[23] Similarly, 87% of animal-bite victims in Mazandaran Province received the VERO vaccine. Geographically, most bites occurred at or near the victim's residence or workplace, predominantly in rural areas—consistent with Nikbakht et al. [23], who reported that 92% of bites occurred in residential or work settings, with half of these occurring in urban Mazandaran. Alian et al.[21] found that 57% of bites occurred in rural areas, whereas Khan et al.[17] reported that more than three-quarters of dog bites occurred on streets, and only 6% took place at home.

In the current study, less than 4% of victims delayed post-exposure prophylaxis (PEP). Overall, 86% completed three doses of the vaccine. Rostampour et al. [16] found a 36% delay in PEP in West Azerbaijan Province between 1391 and 1397, with 77% of victims receiving three doses. Varisli et al. [18] reported a 2% delay, with 25% receiving 3 doses, 21% receiving 4 doses, and 33% receiving 5 doses of the rabies vaccine. Follow-up of animals after biting events in the current study showed that the outcome was unknown in only 14% of cases, compared with 17.5% in Rostampour et al.[16].

The incidence of animal bites remains a major concern, and preventive measures are crucial. In the present study, the annual incidence rates per 100,000 population were 371.39, 503.23, 506.02, and 397.45, respectively. In the systematic review by Shakerian and Sedraei [14], which included 33 studies, the lowest reported incidence was 67 (Mohammadi et al.[9]) and the highest 541 (Babazadeh et al. [6]). Accordingly, the incidence rate in Khoy County was higher than in most Iranian cities. Shakerian and Sedraei [14] also reported a pooled national incidence of 1,200 (!) cases per 100,000 population using meta-analysis, highlighting the substantial burden of this issue. Kiakalayeh et al.[4] reported animal-bite incidence rates in Gilan Province between 2017 and 2022. In 2021, the incidence was 358.3 per 100,000, compared with 371.39 in our study, indicating a higher rate in Khoy. Across all four years (2021–2024), the incidence in Khoy exceeded that reported for Gilan over six years (2017–2021).

The present study conducted a detailed time-series analysis, including testing for mean and variance stationarity, trend analysis, autocorrelation, and partial autocorrelation, and evaluation of 48 model variants. Ultimately, a seasonal model with a 12-month periodicity was selected to forecast animal-bite cases for the next year. Mirzaei et al.[20] performed a similar analysis in East Azerbaijan Province; however, their Fig 3 shows non-stationarity in variance, indicating that variance increased over time, whereas in our study, appropriate transformations achieved stationarity before model fitting.

Some studies, such as Khan et al.[17] in Pakistan, the study did not assess the temporal trend of animal bites. Tuckel and Milczarski (2020) [24] analyzed the time series of dog-bite incidence in the United States from 2005 to 2018. Their results showed that children under nine years old had the highest bite rates throughout the study period. In contrast, in our study, the 10–19 and 40–49 age groups had the highest incidence. In the U.S. data, incidence trends fluctuated (decreasing–increasing–decreasing–increasing–decreasing from 2016 to 2018. Although the time frames differ, our study revealed an increasing trend from 2021 to 2022, followed by a decline in 2024 [24]. Furthermore, throughout the study period, the incidence rate in Khoy was more than twice that reported in the United States.

## Limitations

This study is the first one about animal bites in Khoy County, and reporting incidence rates for 2021–2024 is a strength for the study. Another strength, forecasting new cases for the next 12 months. However, there are some limitations. This study estimated the population of Khoy County using data obtained from the Vice-Chancellor of Health at Khoy University of Medical Sciences and subsequently matched with the Khoy County Governing Office. Then, Incidence rates may be slightly different from the actual rates.

## Conclusion

The annual incidence of animal bites in Khoy County was found to be higher than the national average and greater than that reported in many neighboring cities. Contrary to the common belief that bites occur mainly in rural areas, the study revealed an increasing and concerning trend among urban residents. Dogs accounted for approximately 90% of all bite cases. Given this and the observed urban increase, continuous and well-structured educational and preventive interventions are urgently required in Khoy County. Although treatment and prophylactic procedures for animal bites and rabies are being implemented carefully and promptly—ensuring that all bite victims in the county receive appropriate medical care and prophylaxis according to the national protocol—further preventive measures are necessary to reduce bite incidence.

The temporal trend analysis showed that the peak of animal bites occurs during summer and early autumn. The model predicts that in the upcoming 12 months, the incidence of animal bites will increase until summer, then decrease from autumn through late winter, and that this pattern is likely to persist in future years.

Given the adequate availability of vaccines and related prophylactic resources, it is recommended that policy-makers and health managers prioritize strategies aimed at reducing the incidence of animal bites in this County. Such efforts, particularly through continuous education of high-risk groups, will not only lower vaccine and antibiotic consumption but also reduce human resource demand—ultimately improving efficiency and achieving cost savings in public health programs.

## Supporting information

**S1 Fig. Autocorrelation function (ACF) plot for the SARIMA $(2,0,0)(1,0,0)_{12}$ model.**
(TIF)

**S2 Fig. Partial Autocorrelation function (PACF) plot for the SARIMA $(2,0,0)(1,0,0)_{12}$ model.**
(TIF)

**S3 Fig. Plots are employed to examine assumptions concerning regression residuals: (a) check the normal distribution; (b) show the stability of variance; and (d) assess randomness.**
(TIF)

## Acknowledgments

The authors sincerely thank all colleagues from the Health and Research Deputy Offices of Khoy University of Medical Sciences who provided invaluable assistance throughout the study.

## Author contributions

**Conceptualization:** Parviz Shahmirzalou.

**Data curation:** Parviz Shahmirzalou, Hassan Ebrahimpour Sadagheyani.

**Formal analysis:** Parviz Shahmirzalou.

**Methodology:** Parviz Shahmirzalou, Hassan Ebrahimpour Sadagheyani.

**Resources:** Hassan Ebrahimpour Sadagheyani.

**Writing – original draft:** Parviz Shahmirzalou, Hassan Ebrahimpour Sadagheyani.

**Writing – review & editing:** Parviz Shahmirzalou, Hassan Ebrahimpour Sadagheyani.

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
