## [Decision Letter · Decision Letter 0]

22 Dec 2025

PONE-D-25-60245Epidemiology and Trends of Animal Bites in Khoy County, Northwest Iran: A Retrospective Study (2021–2024)PLOS One

Dear Dr. Ebrahimpour Sadagheyani,

Thank you for submitting your manuscript to PLOS ONE. After careful consideration, we feel that it has merit but does not fully meet PLOS ONE’s publication criteria as it currently stands. Therefore, we invite you to submit a revised version of the manuscript that addresses the points raised during the review process.

Authors need to revise the manuscript as per the comments given in the file.

If applicable, we recommend that you deposit your laboratory protocols in protocols.io to enhance the reproducibility of your results. Protocols.io assigns your protocol its own identifier (DOI) so that it can be cited independently in the future. For instructions see: https://journals.plos.org/plosone/s/submission-guidelines#loc-laboratory-protocols. Additionally, PLOS ONE offers an option for publishing peer-reviewed Lab Protocol articles, which describe protocols hosted on protocols.io. Read more information on sharing protocols at . Additionally, PLOS ONE offers an option for publishing peer-reviewed Lab Protocol articles, which describe protocols hosted on protocols.io. Read more information on sharing protocols at https://plos.org/protocols?utm_medium=editorial-email&utm_source=authorletters&utm_campaign=protocols..

We look forward to receiving your revised manuscript.

Kind regards,

Venu Ramkrishna Shah

Academic Editor

PLOS One

Journal Requirements:

2. In the online submission form, you indicated that “The datasets used and analyzed during the present study are available from the corresponding author upon reasonable request.”

4. We note that Figure 1 in your submission contain map images which may be copyrighted. All PLOS content is published under the Creative Commons Attribution License (CC BY 4.0), which means that the manuscript, images, and Supporting Information files will be freely available online, and any third party is permitted to access, download, copy, distribute, and use these materials in any way, even commercially, with proper attribution. For these reasons, we cannot publish previously copyrighted maps or satellite images created using proprietary data, such as Google software (Google Maps, Street View, and Earth). For more information, see our copyright guidelines: http://journals.plos.org/plosone/s/licenses-and-copyright.

5. We note you have included a table to which you do not refer in the text of your manuscript. Please ensure that you refer to Table 1 in your text; if accepted, production will need this reference to link the reader to the Table.

Additional Editor Comments:

Please revise the article as per the comments added in the file.

Reviewer's Responses to Questions

**Comments to the Author**

1. Is the manuscript technically sound, and do the data support the conclusions?

Reviewer #1: Yes

2. Has the statistical analysis been performed appropriately and rigorously? 

Reviewer #1: Yes

3. Have the authors made all data underlying the findings in their manuscript fully available?

Reviewer #1: Yes

4. Is the manuscript presented in an intelligible fashion and written in standard English?

Reviewer #1: No

5. Review Comments to the Author

Reviewer #1: study is good but some of revisions are required as following (Detailed comments are in manuscript)

1. Language and structural issues - need major revision in grammar, remove duplication of sentences and inconsistency.

2. Introduction: duplication of sentences- require editing

3. Methodology: need clarification and elaboration for study design, data primary or secondary, inclusion criteria, statistical software used.

4. Results: Describe the results coherently and in flow of tables. table 1 need to reorganize, calculated % need to check, duplication of variables.

5. Discussion: could be better and refinement requited.

6. conclusion: could be shorter and specific

7. References: need proper style formatting in all references.

6. PLOS authors have the option to publish the peer review history of their article (what does this mean?). If published, this will include your full peer review and any attached files.). If published, this will include your full peer review and any attached files.

.

Reviewer #1: No

---

## [Author Response · Author response to Decision Letter 1]

19 Feb 2026

Dear Editor-in-Chief, and Referees

Maybe, you are informed about the Internet status in Iran, and the authors apologize for the delay in submitting the revised response.

Sincerely.

---

## [Editor Report · Decision Letter 1]

24 Feb 2026

PONE-D-25-60245R1Epidemiology and Trends of Animal Bites in Khoy County, Northwest Iran: A Retrospective Study (2021–2024)PLOS One

Dear Dr. Ebrahimpour Sadagheyani,

Thank you for submitting your manuscript to PLOS ONE. After careful consideration, we feel that it has merit but does not fully meet PLOS ONE’s publication criteria as it currently stands. Therefore, we invite you to submit a revised version of the manuscript that addresses the points raised during the review process. The following corrections are needed: 1. Table numbers needs to be revised as new table was added, there are two Table 2 in the revised file. please rectify the same in the text too. 2. In figure 5, title can be reframed to make it more clear by adding actual year, rather than mentioning next 12 months.

We look forward to receiving your revised manuscript.

Kind regards,

Venu Ramkrishna Shah

Academic Editor

PLOS One

**Journal Requirements:**

**Additional Editor Comments:**

In the revised manuscript there are two Table 2. authors need to revise the table number and same needs to be quoted correctly in the text for reader's clarity.

---

## [Author Response · Author response to Decision Letter 2]

9 Apr 2026

Dear Reviewers

Please, review attached files and "response to reviewers" file.

Sincerely.

---

## [Editor Report · Decision Letter 2]

12 Apr 2026

Epidemiology and Trends of Animal Bites in Khoy County, Northwest Iran: A Retrospective Study (2021–2024)

PONE-D-25-60245R2

Dear Dr. Ebrahimpour Sadagheyani,

We’re pleased to inform you that your manuscript has been judged scientifically suitable for publication and will be formally accepted for publication once it meets all outstanding technical requirements.

An invoice will be generated when your article is formally accepted. Please note, if your institution has a publishing partnership with PLOS and your article meets the relevant criteria, all or part of your publication costs will be covered. Please make sure your user information is up-to-date by logging into Editorial Manager at Editorial Manager® and clicking the ‘Update My Information' link at the top of the page. For questions related to billing, please contact  and clicking the ‘Update My Information' link at the top of the page. For questions related to billing, please contact billing support..

Kind regards,

Venu Ramkrishna Shah

Academic Editor

PLOS One
---

## [Editor Report · Acceptance letter]

PONE-D-25-60245R2

PLOS One

Dear Dr. Ebrahimpour Sadagheyani,

I'm pleased to inform you that your manuscript has been deemed suitable for publication in PLOS One. Congratulations! Your manuscript is now being handed over to our production team.

Kind regards,

on behalf of

Dr. Venu Ramkrishna Shah

Academic Editor

PLOS One